

# A robust low-level cloud and clutter discrimination method for ground-based millimeter-wavelength cloud radar

Xiaoyu Hu[1], Jinming Ge[1], Jiajing Du[1], Qinghao Li[1], Jianping Huang[1], and Qiang Fu[2]

[1]Key Laboratory for Semi-Arid Climate Change of the Ministry of Education and College of Atmospheric Sciences, Lanzhou
University, Lanzhou, 730000, China.

[2]Department of Atmospheric Sciences, University of Washington, Seattle, WA, 98105, USA.

*Correspondence to*: Jinming Ge (gejm@lzu.edu.cn)

**Abstract.** Low-level clouds play a key role in the energy budget and hydrological cycle of the climate system. The long-term
and accurate observation of low-level clouds is essential for understanding their climate effect and model constraints. Both
ground-based and spaceborne millimeter-wavelength cloud radars can penetrate clouds but the detected low-level clouds are
always contaminated by clutters, which needs to be removed. In this study, we develop an algorithm to accurately separate
low-level clouds from clutters for ground-based cloud radar using multi-dimensional probability distribution functions along
with the Bayesian method. The radar reflectivity, linear depolarization ratio, spectral width and their dependences on the time
of the day, height and season are used as the discriminants. A low pass spatial filter is applied to the Bayesian undecided
classification mask, considering the spatial correlation difference between clouds and clutters. The resulting feature mask
shows a good agreement with lidar detection, which has a high probability of detection rate (98.45%) and a low false alarm
rate (0.37%). This algorithm will be used to reliably detect low-level clouds at the Semi-Arid Climate and Environment
Observatory of Lanzhou University (SACOL) site, to study their climate effect and the interaction with local abundant dust
aerosol in semi-arid region.

## 1. Introduction

Clouds play a crucial role in the Earth-atmosphere system by reflecting solar radiation back to space and trapping
outgoing terrestrial radiation (Bony et al., 2015; Fu et al., 2000, 2018; Quaas et al., 2016). Clouds also produce precipitation

to release large amounts of latent heat into the atmosphere, compensating the atmospheric radiative cooling, which are consequently closely related to the hydrological cycle and global distribution of water resources (Bala et al., 2010; Fu et al.,

2002; Nuijens et al., 2017). Low-level clouds are primarily composed of water droplets and have an overall cooling effect on the climate system. In the context of global warming, tropical low-level cloud amount decreases because of stronger surface turbulent fluxes and dryer planetary boundary layer, generating a positive climate feedback through a reduction in the reflection of short-wave radiation (Brient and Bony, 2012; Zhang et al., 2018); While the liquid water path of low-level clouds over mid- to high-latitude tends to increase due to a reduced conversion efficiencies of liquid water to ice and precipitation, which leads

to a negative feedback (Ceppi et al., 2016; Terai et al., 2016). However, the magnitude of these low-level cloud feedbacks responds inconsistently in different climate models, producing a wide range of equilibrium climate sensitivity (Watanabe et al., 2018; Zelinka et al., 2020; Mace and Berry, 2017). To reduce this uncertainty, accurate long-term observations are important to characterize low-level clouds and understand their climate feedbacks (Turner et al., 2007; Garrett and Zhao, 2013; Toll et al., 2019).

The ground-based cloud radars can probe the vertical structure of low-level clouds in high temporal-vertical resolution, including multi-layer clouds (Kim et al., 2011; van der Linden et al., 2015). Due to substantial progress in the development and application of ground-based radars, there are increasing numbers of ground-based millimeter cloud radars being deployed all over the word (Arulraj and Barros, 2017; Kollias et al., 2019; Huo et al., 2020). Their short wavelengths allow the radars to detect clouds with small droplets and infer the microphysical and dynamical cloud processes (Kollias et al., 2007a). A Ka-

band zenith radar (KAZR) has been continuously running at the Semi-Arid Climate and Environment Observatory of Lanzhou University (SACOL) since 2013 (Huang et al., 2008b; Ge et al., 2018, 2019) to investigate cloud properties over the site. SACOL is located in the downwind dust transport path about 2000 km to the east of the Taklimakan Desert (i.e. one of the most important global sources of atmospheric dust) (Su et al., 2008; Huang et al., 2007; Ge et al., 2014). Low-level clouds in this semi-arid region with abound dust aerosols acting as cloud condensation nuclei may contain a larger number of small

droplets (Givati and Rosenfeld, 2004; Huang et al., 2006), which may reflect more short-wave radiation, merge more slowly

to fall as precipitation (Xue et al., 2008; Huang et al., 2014), and thus affect regional energy budget and water cycle in a

specific way. Therefore cloud observations are vital to understand their effects on the local fragile dryland ecosystem (Fu and

Feng, 2014; Huang et al., 2017, 2018, 2020). MMCR-observed cloud echoes in the lowest 3 km above ground level (AGL)

are often contaminated by unwanted clutters, mostly insects for midlatitude continent (Clothiaux et al., 2000), presenting non-

Rayleigh scattering at millimeter wavelength with their large physical size, which need to be removed for the low-level cloud

research.

The most distinctive feature of clutter is the non-spherical shape, which causes larger linear depolarization ratio (LDR)

values than that of low-level cloud droplets. Thus, a threshold of LDR can be used to separate clutters from cloud droplets.

For instance, Görsdorf et al. (2015) chose a threshold of $-20$ dB in German, Zheng et al. (2016) used $-16$ dB in Tibetan Plateau,

and Oh et al. (2018) took $-15$ dB in Korea to achieve the purpose. Although the simple threshold can remove a large part of

the clutters, not all the radar range bins with high LDR are necessarily clutters. For example, the non-spherical melting

hydrometeors also generate a significant LDR peak in the melting layer (Kowalewski and Peters, 2010). Furthermore, the

threshold fails to separate clutters from hydrometeors when their LDR probability density function (PDF) curves are in the

overlapping area. Instead of a single LDR threshold, using more attributes to build multi-dimensional PDFs can adequately

describe the different properties of clouds and clutters in multi-dimensional space, thereby decrease the overlapping region

and reduce the fraction of ambiguous classifications. For instance, Golbon-Haghighi et al. (2016) used three-dimensional PDFs

and two-days training data to successfully identify fixed clutters such as buildings and trees for weather radar. The latest

CALIPSO cloud aerosol discrimination algorithm uses five different parameters to build multi-dimensional PDFs and

improves the previous classifications (Liu et al., 2019). However, samples are more scattered in the higher-dimensional space

and are less likely to capture the characteristics of various insect clutters, for examples, which have unique yet complicated

behaviors, using short-term data. To clearly characterize the insect's behaviors, a large amount of long-term training data is

required to build an accurate multi-dimensional PDF for such clutters.

In this study, we develop a robust algorithm to distinguish low-level clouds from clutters. We first remove the

background noise, precipitation and melting layer from radar measurement. We then examine cloud radar observations and

select discriminants using radar reflectivity, LDR, and spectral width (SW). Next, we utilize one-year micro pulse lidar (MPL)

data to establish the multi-dimensional PDFs for clouds and clutters by noting that lidar is not susceptible to clutters and

therefore can provide accurate cloud base measurements. The obtained PDFs are used to train the Bayesian classifier which is

able to determine whether a radar range gate is cloud or clutter, by comparing their estimated probabilities. Finally, a low pass

time-spatial filter is applied to the radar range gates where the Bayesian classifier does not work. Section 2 illustrates radar

and lidar observations. The details of the algorithm are described in Sect. 3. Using the presented method, in Sect. 4, several

case studies and one-year evaluation are showed. Finally, the summary and discussion are provided in Sect. 5.

## 2.    Instruments and datasets

The KAZR at SACOL site (35.57° N, 104.08° E) is a zenith-pointing dual-polarization cloud radar operating at 35 GHz.

It uses an extended interaction Klystron (EIK) amplifier with a peak power of 2.2 kW. KAZR has a narrow (0.3°) antenna

bandwidth and high temporal (4.27 s) and vertical (30 m) resolutions. The cloud radar has been running continuously since

2013 and provides radar reflectivity, doppler vertical velocity, and SW in each radar range gate from 0.9 km to 17.6 km AGL.

The LDR is derived as the ratio of cross-polarized reflectivity to co-polarized reflectivity. More details about the KAZR are

described in (Ge et al., 2017). In this study, we use radar reflectivity, LDR and SW as discriminants to separate low-level

clouds and clutters. The vertical velocity is also used to identity precipitation and melting layer to reduce the potential

misclassification. A micro pules lidar (MPL), working at 527 nm wavelength with 1-min temporal and 30-m vertical resolution,

is simultaneously running near by the KAZR (Huang et al., 2008a; Xie et al., 2017; Xin et al., 2019). Since lidar is not

susceptible to the clutters, the lidar-measured cloud base is accurate, which can be used to establish dependable multi-

dimensional PDFs for both clouds and clutters. We use one-year lidar data (August 2014 to July 2015) to build the multi-

dimensional PDFs to train the Bayesian classifier (in Sect. 3.2), and another year data (August 2013 to July 2014) to evaluate

the algorithm (in Sect. 4.2). We choose the later year to build the PDFs because there are more observations available in that

year, which helps build reliable PDFs.

## 3.  Low-level cloud and clutter discrimination algorithm

The algorithm uses radar-observed variables to describe the different characteristics of clouds and clutters. A probability of a radar range gate to be a cloud or clutter is estimated based on the Bayesian method using the pre-established multi-dimensional PDFs. The step-by-step procedure of the algorithm is summarized in Figure 1. Before constructing multi-dimensional PDFs of cloud and clutter, the radar echoes including background noise, precipitation and melting layer need to be removed from radar measurement (Sect. 3.1). We then use the simultaneous lidar measurement to distinguish clouds and clutters (Sect. 3.2). Any radar echoes above the lidar cloud base height are considered to be clouds, and the below are clutters. After the multi-dimensional PDFs are created, the Bayesian method is used to estimate the probability of any given radar observation being cloud or clutter (Sect. 3.3). Although the multi-dimensional PDFs do provide more comprehensive description of the difference, the Bayesian classifier can only discriminate cloud from clutter when all radar discriminants (radar reflectivity, LDR and SW) are given. The fact that LDR measurement can merely be derived when both co- and cross-polarized reflectivities are available, causes non-negligible amount of undecided classification. A final time-spatial filter is therefore used to identify these radar range gates, considering that clouds are more spatially correlated than clutter (Sect. 3.4).

### 3.1.  Removing noise and non-cloud meteorological target

The radar background noise is firstly removed using the noise equivalent reflectivity (NER) (Kalapureddy et al., 2018), that is $r^2 \times Z_{start\ range}$, where $r$ is height and $Z_{start\ range}$ is the noise equivalent reflectivity of the first range gate from the bottom. Here we use a $Z_{start\ range}$ of $-60 dBZ$, because it fits the radar noise level well after several trials. Figure 2 shows an example of raw and noise-removed reflectivity from local time 12:08 to 12:29 on May 28th, 2014. The reflectivity is irregularly dispersed below 2.6 km, which is caused by flying insects, while it is distributed more homogeneously inside the cloud layers above 2.6 km (Figure 2a). This is because clutter reflectivity is determined by the size and number of individual

insects in a radar range gate and is little relevant to its surrounding insects. But the reflectivity inside a cloud is largely

controlled by environmental variables which is highly spatially correlated. The NER curve (blue dashed line in Figure 2b) fits

well with the background noise, and almost all the background noise is removed (Figure 2c). Additionally, the flat cloud

boundary around 4.5 km, the fluctuant cloud boundary that may be caused by gravity wave around 6.4 km, and the broken thin

cirrus boundary around 9.2 km are all kept (Figure 2a and c). It is obvious that the clutter reflectivity is not necessarily less

than the cloud reflectivity (Figure 2b). A single threshold of reflectivity cannot adequately separate clouds from clutters, and

therefore multi-dimensional PDFs are needed to describe their differences.

The non-cloud meteorological targets in the low-level atmosphere, such as precipitation and melting layer, usually have

different features from cloud droplets. If we put them into the cloud category, it would affect the accuracy of the created PDFs

to characterize clouds and clutters. Thus, these non-cloud meteorological targets need to be removed before establishing the

multi-dimensional PDFs. Rain drops are normally larger than cloud droplets and have fast fall velocity, thus radar reflectivity

and vertical velocity can be used to identify precipitation (Shupe, 2007). In some cases, the radar-measured velocity may be

erroneously aliased (Kollias et al., 2007b; Zheng et al., 2017) when the naturally occurring velocity is larger than the maximum

unambiguous velocity ($V_{max}$, ±10.38 m/s for KAZR at SACOL), as shown in Figure 3. From this heavy precipitation event,

one can see that the radar reflectivity is attenuated above 3 km (Figure 3a). The velocity aliasing happens at the lower level of

atmosphere where radar measured velocity suddenly reverses from large downwards to large upwards (harsh red area in Figure

3b and blue dots near the right gray line in Figure 3d). The absolute value of the gate-to-gate velocity difference is used to

check if velocity is aliased. For aliased velocity, that is when absolute velocity difference exceeds $1.5 \times V_{max}$, $2 \times V_{max}$ is

subtracted from (or added to) the aliased velocity if the velocity difference is positive (or negative) (Johnson et al., 2017; Sokol

et al., 2018). The adjusted velocity is shown in Figure 3c, where the upwards velocity at the lower level of atmosphere is de-

aliased to downwards (smooth blue region in Figure 3c and orange dots in Figure 3d). The de-aliased velocity and reflectivity

are then averaged over 1 minute to reduce the effect of wind drift effects to identify precipitation. These range bins with

averaged reflectivity greater than −10 dBZ and averaged velocity lesser than −3 m/s are identified as precipitation (Chandra et

al., 2015). However, the drizzle with smaller sizes and lower velocity (O'Connor et al., 2005; Kollias et al., 2011) may not be

identified by the above method. Thus, the radar echoes that below the lidar detected cloud base, while still being connected to

the cloud, are marked as drizzle (Wu et al., 2015; Yang et al., 2018), and removed from the training data.

Water coating ice particles inside the melting layer are largely non-spherical, therefore have high LDR values, similar

with insects (Islam et al., 2012; Brandes and Ikeda, 2004). This can be seen from Figure 4c. The melting layer at around 2.8

km has relatively higher LDRs than the precipitation below and ice particles above. Clutters near the surface before the

precipitation reaching the surface at about 20:30 have similar high LDR values. Clutter layer can appear as high as 3 km AGL

during daytime in warm season at SACOL site, which is close to or even higher than melting layer height. In order to avoid

wrongly identifying the melting layer with high LDR as clutters, the melting layer is recognized by analyzing the gradient of

reflectivity and velocity that has a large value associated with the melting layer (Matrosov et al., 2007; Baldini and Gorgucci,

2006; Perry et al., 2017). The peak of reflectivity' × velocity' (Figure 4e) is located as the middle of melting layer for each

identified precipitation profile, then the height of maximum (reflectivity' × velocity')' up to 500 m above (below) the peak are

defined as the top (bottom) of melting layer as shown in Figure 4e with red dots (Khanal et al., 2019; Devisetty et al., 2019).

The identified melting layer and precipitation are plotted in Figure 4a-c as black dots and gray shading area.

### 3.2. Creating multi-dimensional PDFs

To capture the differences between clouds and clutters as accurately as possible, we need to choose the appropriate

discriminants before creating the PDFs for both. From a statistical point of view, the description of differences in higher-

dimensional space is generally more complete than in lower-dimensional space. Increasing the number of discriminants could

decrease the overlapping region of the two PDFs, thereby reducing the fraction of ambiguous classifications (Liu et al., 2004).

However, only when the added discriminant is largely independent of the other used, can it improve the classification

significantly (Liu et al., 2009). After carefully examining all radar variables for many specific clutter and cloud cases, we chose

radar reflectivity, LDR, SW along with their time-height and seasonal dependence as discriminants. LDR is chosen because it

has discrepant distributions for cloud and clutter due to their shape difference (cloud droplets are largely spherical while clutters

are non-spherical). Insects' number density and sizes make them often generate low radar reflectivity, which has a similar

range with strati and broken cumuli (Luke et al., 2008), but is commonly higher in warm seasons when they swarm (Abrol,

2015). The seasonal dependence of radar reflectivity is considered as a factor to build the PDFs. Clutters also generally have

lower SW and lower vertical velocity because insects may actively oppose environmental vertical motion and control their

own flying behavior, while cloud particles are more vulnerable to small-scale local turbulence and entrainment processes

(Geerts and Miao, 2005). Yet after checking both variables, we found that distributions of SW for clouds and clutters are more

discrepant than that of vertical velocity, thus SW is used to build the PDFs rather than using vertical velocity directly. One

distinctive character of insect that differs from other fixed clutter is that their behaviors are influenced by many natural factors

(Thomas et al., 2003; Johnson et al., 2016; Chapman et al., 2015). For example, insects' number density has a high correlation

with surface temperature (Luke et al., 2008), thus the maximum height and radar echo intensity of insects have strong diurnal

cycles (Wood et al., 2009; Hubbert et al., 2018). The time and height variations of radar echoes are thereby considered in the

construction of multi-dimensional PDFs.

Once the discriminant factors are selected, the cloud and clutter samples need to be extracted for building the multi-

dimensional PDFs. The radar echo above the lidar cloud base height after removing noise and non-cloud meteorological targets

are considered to be clouds, otherwise are clutters. Based on the lidar auxiliary data, all the radar echoes below 3.6 km from

August 2014 to July 2015 are separated into cloud or clutter samples. Figs. 5 and 6 show the multi-dimensional PDFs for

different local time and heights for warm and cold seasons, respectively. After examining one-year data, it is found that 3.6 km

AGL is the highest level that clutters can reach at the SACOL site. As expected, clutters tend to have lower reflectivity (lower

density), larger LDR (non-spherical shape) and lower SW (less turbulent motion) compared with cloud (Figure 5 and Figure

6). Insect activities are largely influenced by temperature, thus the clutter appears mostly during daytime and its height has an

obvious diurnal cycle. It is also notable that there are no clutters above 2.7 km during the nighttime (Figure 5c1 and d1, Figure

6c1 and d1). The three radar variables for cloud and clutter still have contrasting distributions during cold season. Nevertheless





both cloud and clutter occur less frequently compared to warm season (Zhu et al., 2017). Note that some overlapping regions

of cloud and clutter PDFs still occur (e.g. Figure 5b3). However, the multi-dimensional PDFs made the ambiguity area much

smaller compared with the results by only using a single discriminant. The significant differences between clutter and cloud

PDFs (Figs. 5 and 6) can be used to adequately separate them more accurately.

### 3.3. Generating classification mask based on Bayesian method

The obtained multi-dimension PDFs are then used to train the optimal Bayesian classifier to separate clouds and clutters

for any observed discriminants ($X^O$). According to Bayesian method, the probability of a radar range gate with discriminants

$X = X^O = Reflectivity^O, LDR^O, SW^O, Time^O, Height^O, Season^O$ being class $C_i, (i \in [cloud, clutter])$ can be estimated

as:

$$p(C_i|X = X^O) = \frac{p(X = X^O|C_i)p(C_i)}{p(X = X^O)} \tag{1}$$

where the priori probabilities are assumed to be equal for all classes (Ma et al., 2019; Golbon-Haghighi et al., 2016), which

means $p(C_{cloud}) = p(C_{clutter}) = 1/2$. Furthermore, as $p(X = X^O)$ is the same for all classes, hence Eq. (1) can be rewritten

as

$$p(C_i|X = X^O) = Kp(X = X^O|C_i) \tag{2}$$

where $K$ is constant for all classes

$$K = \frac{1}{2p(X = X^O)} \tag{3}$$

and $p(X = X^O|C_i)$ is the conditional probability of discriminants being $X^O$ for each class, which has been derived from one-

year training data as descript in Sect. 3.2.

For any given observation of discriminants, the posterior probability for each class $p(C_i|X = X^O)$ is estimated

accordingly and compared to decide its category. The radar range gate belongs to cloud only when $p(C_{cloud}|X = X^O)$ is larger

than $p(C_{clutter}|X = X^O)$. And vice versa, if $p(C_{clutter}|X = X^O)$ is larger than $p(C_{cloud}|X = X^O)$, it is considered to be a

clutter gate.

Figure 7e, f and g show an example of estimated probabilities from Bayesian classifier, and classification mask from local time 05:00 to 22:00 on September 24th, 2013. Unsurprisingly, these radar range bins with low reflectivity (Figure 7a), high LDR (Figure 7c), and low SW (Figure 7d) are considered more likely to be clutters rather than clouds (Figure 7e, f and

g), while the high reflectivity, low LDR and high SW correspond with higher probability of being clouds (Figure 7a-g). When the individual three radar variables disagree on the classification, for example, these clutter from 12:00 to 16:00 near the surface with high reflectivity and high SW (likely to be clouds) and high LDR (also likely to be clutter), the Bayesian classifier can still correctly separate them as shown in Figure 7g. Note that the green dots in Figure 7g represents undecided classification mask by Bayesian classifier, which may be caused by the two equal probabilities, but more likely, the absence of either. The

undecided mask will be discussed in Sect. 3.4.

### 3.4. Applying low pass spatial filter to undecided mask

Bayesian classifier is able to separate clouds and clutter in most cases when all the radar discriminants as described in Sect. 3.2 and 3.3 are offered. However, the cloud radar may not always provide valid observations. For example, LDR can only be computed when both co- and cross-polarized reflectivities are available. Figure 7a and b show the reflectivities of co-

and the cross-polarized channels, respectively. There are some range gates where co-polarized reflectivity detected signal (cloud or clutter) while no signal detected in cross-polarized channel, which causes the missing LDR in these radar range gates (e.g., the rightmost range bins above the lidar cloud base and some bins scattering near-surface in Figure 7c). Without the LDR input data, Bayesian classifier fails to work (green dots in Figure 7g), because no conditional probability was established for an incomplete $X^O$. Mathematically, there are several approaches to deal with missing data for Bayesian method, such as

assuming a distribution of them (Linero and Daniels, 2018). However, in practice, we find it is uneconomical to solve such issue. Rather, we utilize the spatial correlation difference between clouds and clutters to process the Bayesian undecided classifications, which is more effective and simpler. As mentioned earlier, cloud droplets are highly correlated in time and space, while clutters do not have the same feature. For those radar bins cannot be identified as cloud or clutter from the

probability estimate, we use the neighboring range gates to provide information to help make the final decision. A spatial filter

with five range bins in vertical and five range bins in the horizontal, which is centered at each undecided classification bin, is

employed here (Marchand et al., 2008; Hu et al., 2020). Following Ge et al. (2017), if the number of cloud range bins in the

box is less than 13, this range bin is considered to be clutter, otherwise it will be marked as a cloud bin. The final classification

mask result is shown in Figure 7h. Comparing with lidar observation on the same day, the undecided range bins are correctly

categorized into clutters (green dots turned to brown), and clouds (green dots turned to blue above lidar cloud base) after

applying the low pass spatial filter. It is clear from Fig. 7h that clutter layer height has an apparent diurnal cycle and the insects'

number density is much stronger in the early afternoon near the surface (patchy high reflectivity rather dotted low reflectivity).

This is why time and height are also chosen as the discriminants.

## 4.    Result

### 4.1.  Case study

We apply the identification algorithm to a whole year of radar data to discriminate low-level cloud and clutter

discrimination. The results are compared with the simultaneous lidar cloud base to demonstrate the performance of the

algorithm.

Figure 8 shows a case of broken cumulus from local time 16:27 to 17:30 on April 15[th], 2014. During this period, a

substantial presence of insects is observed below the broken cumulus. The top of the insect layer is around 1.6 km, where is

also the cloud base height observed by lidar and detected by our algorithm (Figure 8d). From the radar reflectivity image in

Figure 8a, the cloud droplets begin to dissipate due to entrainment (Chernykh et al., 2001; Pinsky and Khain, 2019) and have

similar reflectivity values as clutters (around −50 dBZ) around cloud base. As shown in Figure 8b, clutters have the LDRs

mostly greater than −15 dB but cloud has relatively smaller LDR values. The high SW above the cloud base (more than 0.4

$m^2/s^2$) indicates strong turbulence inside the cumulus. Combining all these radar variables together, our clutter identification

algorithm shows a great agreement with lidar detection (Figure 8d).



Figure 9 shows a case of stratus clouds embedded in insect layers. The reflectivity inside cloud is similar to the clutter reflectivity (between −40 to −20 dBZ), but is distributed more homogeneous in time and space (Figure 9a). Note that for these flat clouds, Kalapureddy et al. (2018) used the standard deviation of reflectivity to remove clutter. However, this method causes some false positives (cloud is wrongly identified as clutter) around fuzzy cloud edges. The stratus cloud typically is more
featureless than cumulus (Fig. 8) due to the absence of active convective elements (Harrison et al., 2017), and it has lower SW values which may fall within the same range as clutter (below 0.4 m²/s², Figure 9c). Thus, in this case, the LDR (Figure 9b) and spatial filter in our method made the major contribution to separate them.

Figure 10 shows a case of precipitating stratocumulus. The drizzle droplets that fall from the cloud base are kept as clouds (Figure 10d), since they have relatively small falling velocity and reflectivity, and cannot be recognized as precipitation by the
algorithm. The edge between clutter and drizzle are blurry in radar reflectivity and SW (Figure 10a and c). Under this circumstance, the algorithm identifies the clutter near the surface with large LDR (larger than −15 dB), but keeps the drizzle as hydrometeors with low pass filter since they are temporal and spatial correlated (Figure 10b). Note that although the bottom of identified hydrometeors is coincidental with the top height of clutter layer (Figure 10d), it does not mean that the drizzle droplets "suddenly" all evaporate when they fall into the insect layer. The drizzle may still fall toward the ground, however
the signals are much smaller than that from the insect layer. In other words, the clutter mask does not necessarily mean only clutter being in this range bin, rather the backscattered power is largely dominated by insects.

Figure 11 shows a case of shallow cumulus near the surface in cold season. Compared with the earlier three cases (Figure 8-10), the clutter in this case is less organized. There is no dense insect layer gathering near the surface. The different behaviors of insects in warm and cold season (Figure 8-10 vs Figure 11) are why seasonal variation is chosen as a discriminant. The radar
reflectivity in the cumulus is more homogenous than the scattering clutter (Figure 11a) and can easily be identified even though human eyes. Shallow cumulus has LDR less than −20 dB whereas clutter has higher LDR greater than −15 dB (Figure 11b). Higher SW values (around 0.3 m²/s², Figure 11c) in the cumulus indicate that the cloud droplets are more affected by small-scale local turbulence and entrainment processes. The algorithm is able to screen out the shallow cumulus in cold season and



filter out the clutter (Figure 11e).

**4.2. One-year evaluation**

To further objectively demonstrate the performance of this algorithm, probability of detection ($P_D$) and false alarm rate ($P_{FA}$) are calculated using one-year data (August 2013 to July 2014) that are defined as:

$$P_D = \frac{TP}{TP + FN}, \qquad P_{FA} = \frac{FP}{FP + TN} \tag{4}$$

where the number of $TP$ (true positives), $TN$ (true negatives), $FP$ (false positives) and $FN$ (false negatives) are based on our algorithm identified clutter ("positive" of "negative") validated by lidar detection ("true" or "false" of clutter classification mask). Note that the evaluation is focused on the identified clutters rather than low-level clouds, because lidar power is often attenuated by optically thick low-level water clouds, leading to a significant discrepancy between radar- and lidar-measured low-level clouds, while the "true" or "false" of clutter only relies on lidar cloud base height, which would cause less uncertain in the assessment.

Figure 12 illustrates the $P_D$ and $P_{FA}$ as functions of reflectivity (a), LDR (b), SW (c), time (d) and height (e). The $P_D$ (solid lines) is usually above 98%, except when reflectivity is larger than −10 dBZ (Figure 12a), LDR lower than −15 dB (Figure 12b), or SW larger than 0.2 m²/s² (Figure 12c), where clutters have similar properties as clouds. From Figure 5 and Figure 6, one can see that these are only small portions of the data. So the seasonally- and yearly-averaged $P_D$ are all above 98% (Figure 12f). Similarly, for the cloud with reflectivity lower than −30 dBZ (Figure 12a), LDR larger than −20 dB (Figure 12b), and SW lower than 0.1 m²/s² (Figure 12c), there are chances that clouds are falsely identified as clutters (higher $P_{FA}$, dashed lines). The $P_{FA}$ are below 0.5% in all seasons (Figure 12f). Using a single LDR threshold to filter out clutter would induce a sharp increase of $P_D$ from 0% to 100% at the threshold point. Very different from that, by using multi-dimensional PDFs with the Bayesian method, it can correctly identify cloud-like clutter and clutter-like cloud, thus increase the accuracy of the classification mask. Both $P_D$ and $P_{FA}$ are less fluctuating with time (Figure 12d) and height (Figure 12e) compared with the three radar variables (Figure 12a-c), except for $P_D$ above 3.5 km, where the clutter is extremely rare (fewer samples).

This indicates that the time and height variations of cloud and clutter features are well captured by the multi-dimensional PDFs. The $P_D$ and $P_{FA}$ of whole year (black lines) are more consistent with that of warm season (red line), because clutters are more frequently appear in warm season. Overall, the one-year evaluation shows that the algorithm can successfully filter clutter out with a high value of $P_D$ (98.45%) and a very low value of $P_{FA}$ (0.37%) as shown in Figure 12f.

## 5.    Summary and discussion

We develop a low-level cloud and clutter discrimination algorithm for a ground-based cloud radar based on multi-dimensional PDFs with the Bayesian method using cloud radar reflectivity, LDR, SW and their time of the day, height and season dependences as discriminants. A low pass spatial filter is applied to the Bayesian undecided classification mask, considering the spatial correlation difference between clouds and clutters. The case studies indicate the algorithm can filter out most of the clutter while still maintaining the low-level clouds (including drizzle), even when they are embedded in clutter layer. Unlike the traditional way by selecting a single LDR threshold to remove the clutter, this algorithm particularly shows higher accuracy for clutter-like cloud or cloud-like clutter. The one-year evaluation demonstrates a good performance of this algorithm (98.5% detection rate and 0.4% false alarm rate). For the quantitative evaluation, the lidar detected cloud base is assumed to be perfectly correct, and the small temporal and spatial offsets between the radar and lidar are assumed to have small impact. We conclude that this algorithm satisfactorily retains low-level clouds and removes radar clutter at SACOL site.

For the non-cloud low-level meteorological target, such as precipitation and melting layer, we use radar observation itself to identify them (Chandra et al., 2015; Matrosov et al., 2007). Although it might not be as reliable as the method by combining the radar with other instruments such as rain gauge, it would still be enough to effectively reduce the misclassification of clutter and clouds. The more accurate estimation of rain rate will be carried out in our future work, along with this algorithm, used to provide more reliable low-level cloud and precipitation radar data to study its climate effect and the interaction with local abundant dust aerosol in semi-arid region.

**Data availability**

Both the lidar and radar data used in this study can be acquired from the SACOL site (http://climate.lzu.edu.cn).

**Author contributions.**

XH and JG designed the study. XH, JD and QL performed the cloud and clutter discrimination. XH and JG prepared the

manuscript with significant contributions from all co-authors.

**Competing interests**

The authors declare that they have no conflict of interest.

**Acknowledgments**

This work was supported by the National Science Foundation of China (41922032, 91937302, 41875028) and the

National Key R & D Program of China (2016YFC0401003). We also would like to thank the SACOL team

(http://climate.lzu.edu.cn) for supporting the radar and lidar data.

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



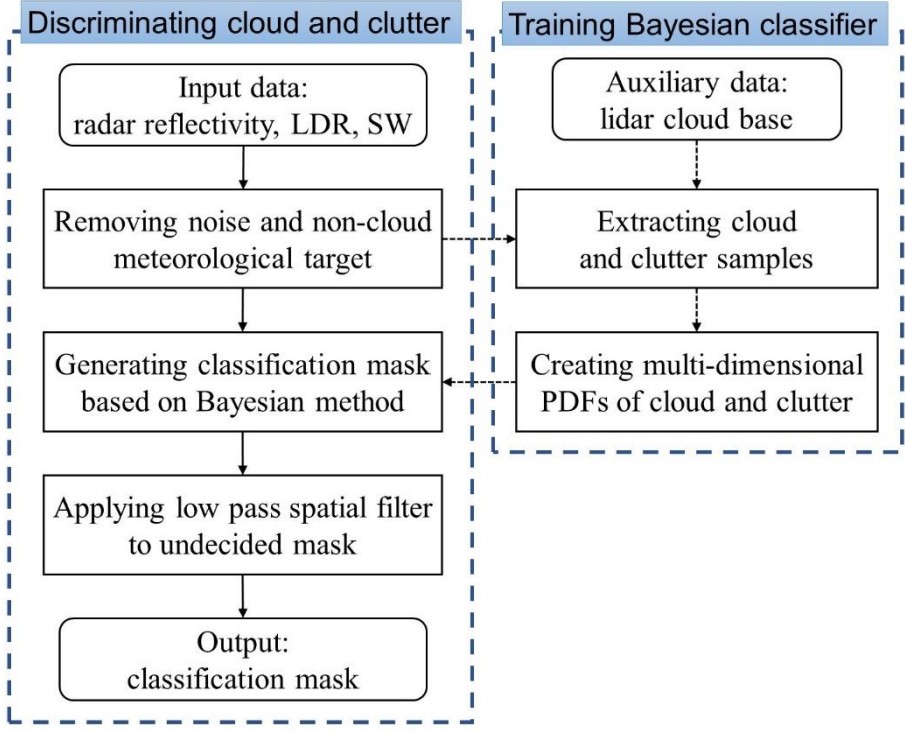

Figure 1. Schematic flow diagram for cloud and clutter discrimination. The right panel (connected by dashed arrow) is only

executed once to train the Bayesian classifier.

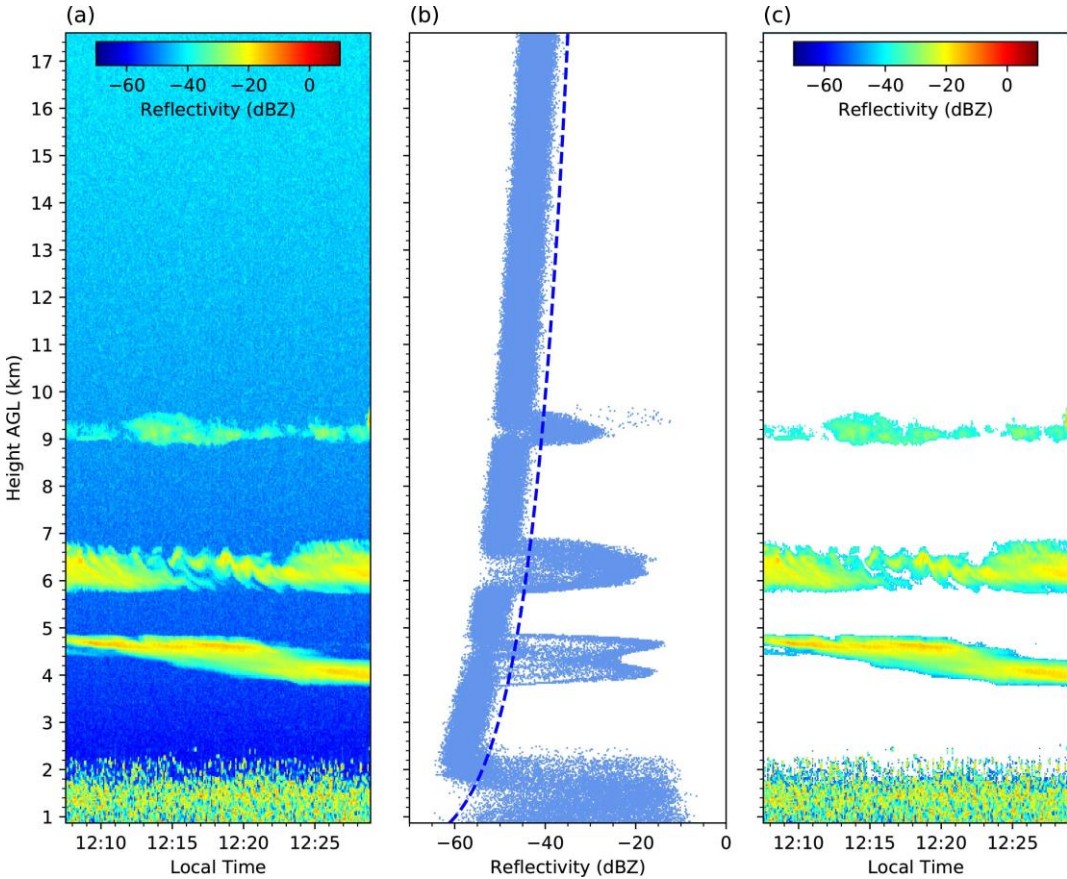

Figure 2. (a) Raw reflectivity and (c) noise-removed reflectivity from local time 12:08 to 12:29 on May 28th, 2014. (b) 300

reflectivity profiles of the same duration, the blue dashed line is noise equivalent reflectivity curve.



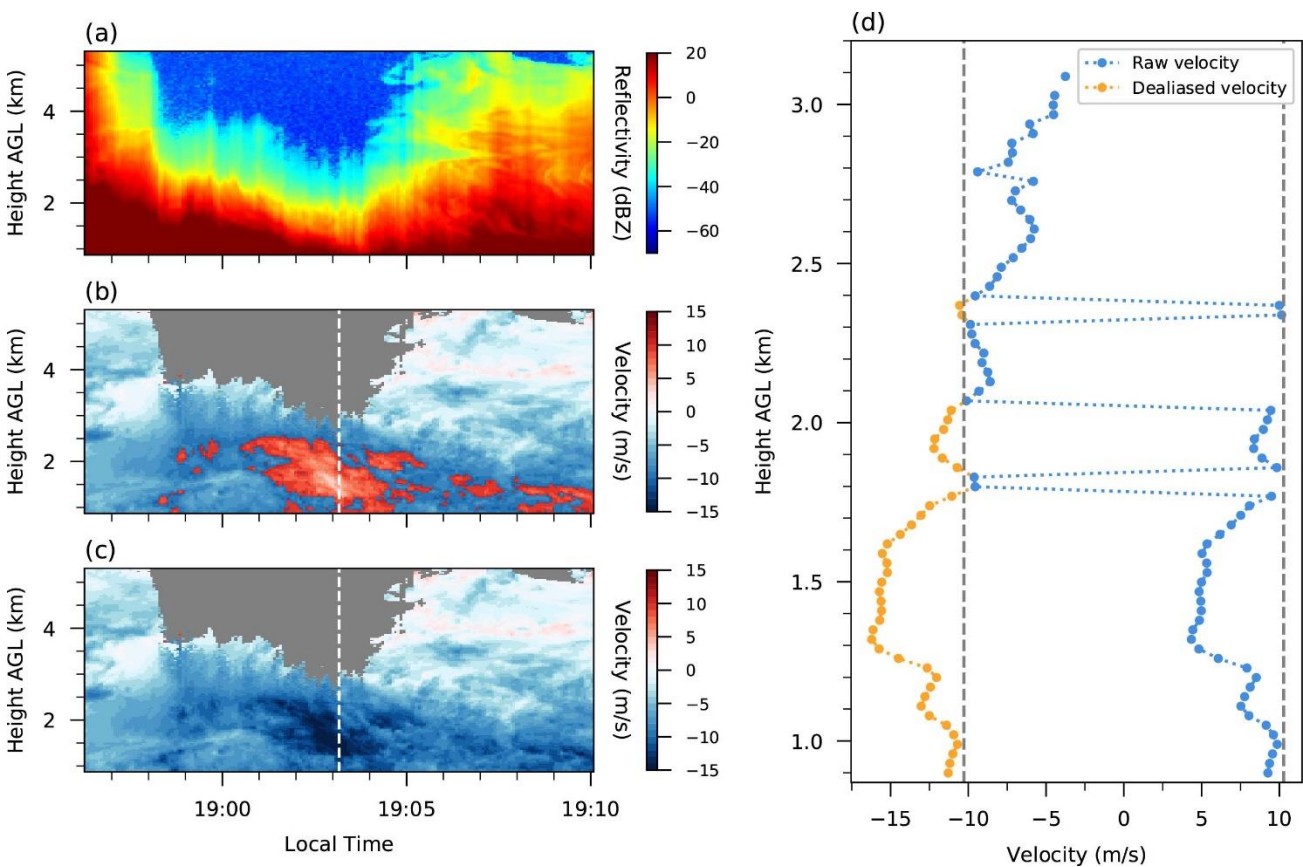

Figure 3. (a) Reflectivity, (b) radar measured doppler velocity and (c) de-aliased velocity from local time 18:56 to 19:10 on

August 30th, 2013. (d) Raw and de-aliased velocity profile of the white dashed line in left panels, the gray dashed line is the

maximum unambiguous velocity (±10.38 m/s for SACOL KAZR). Positive velocity represents upwards velocity.



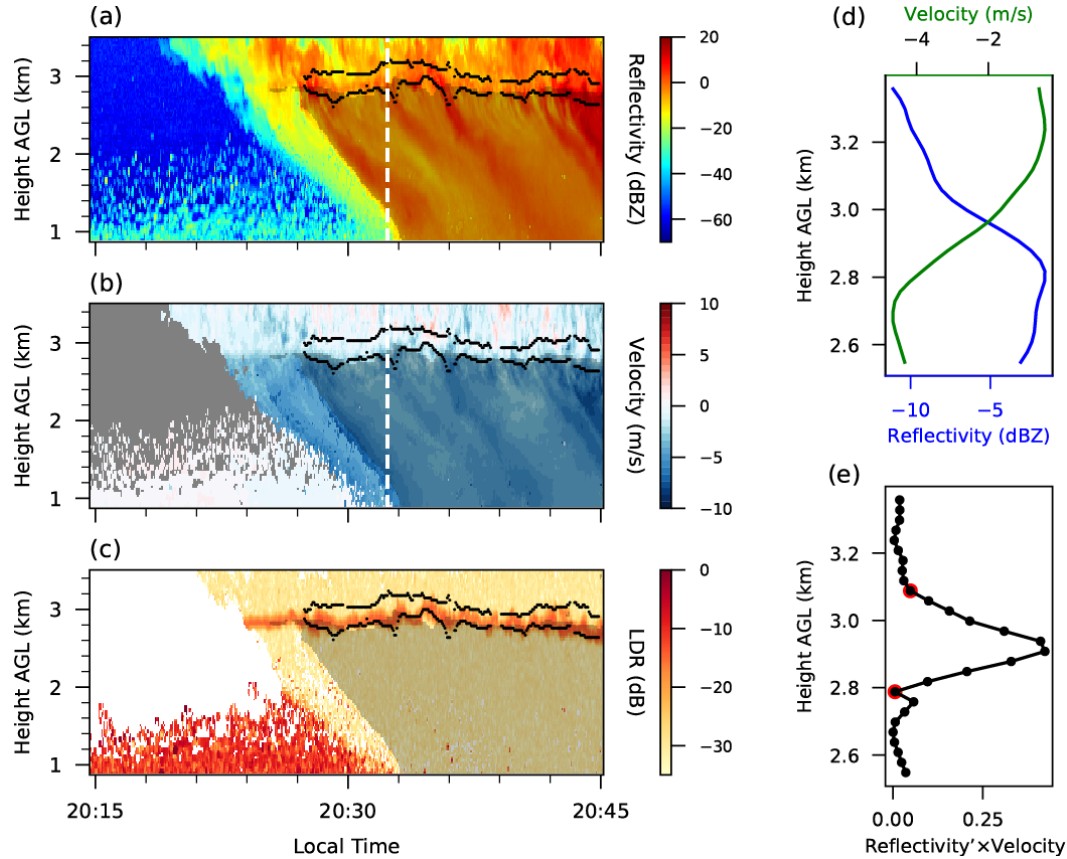

Figure 4. (a) Reflectivity, (b) velocity, and (c) LDR from local time 20:15 to 20:45 on August 10th, 2013. (d) Reflectivity and velocity, and (e) reflectivity' × velocity' profile of the white dashed line in left panels. Black dos and gray shading area in left panels are identified melting layer and precipitation. Red dots in (e) are the identified bottom and top of melting layer.





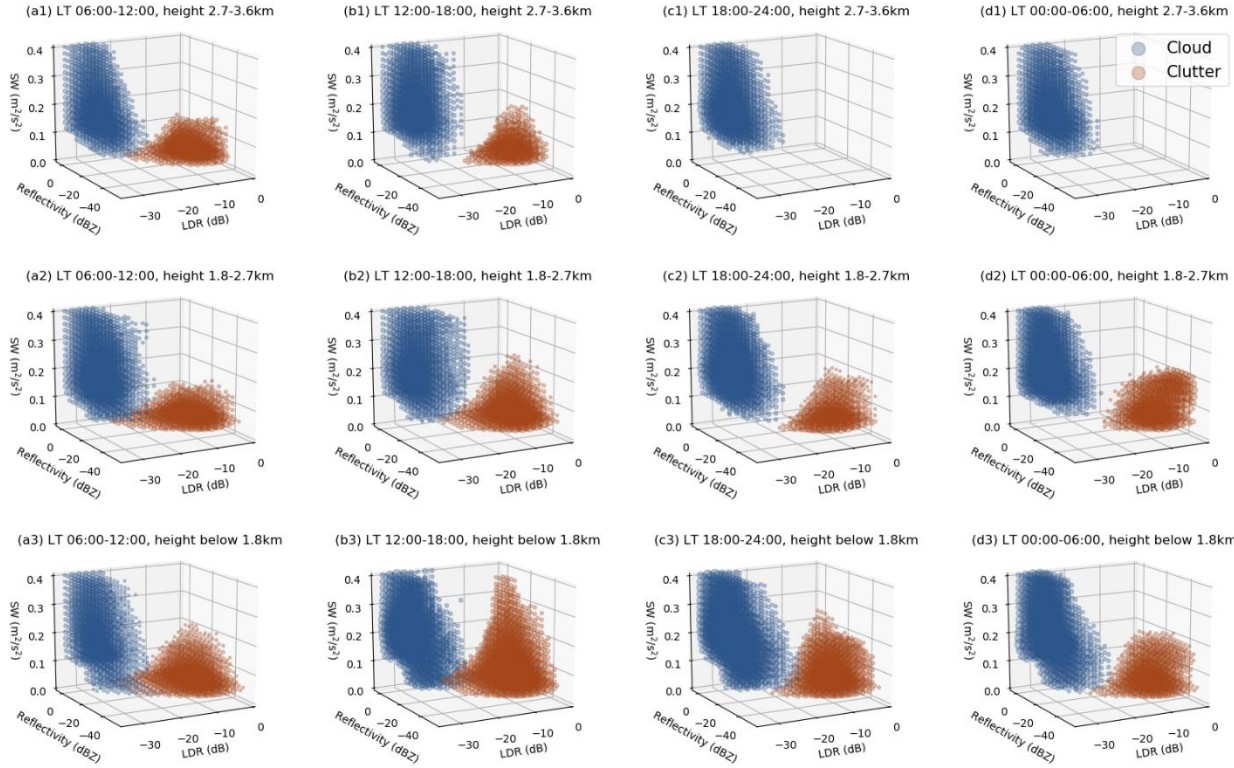

Figure 5. The multi-dimensional PDFs of clutters (brown dots) and cloud droplets (blue dots) at local time 06:00-12:00 (column a), 12:00-18:00 (column b), 18:00-24:00 (column c) and 00:00-06:00 (column d), and height below 1.8 km (row 3), 1.8-2.7 km (row 2) and 2.7-3.6 km (row 1) in warm season (April to September). The size of dots represents the value of probability density.





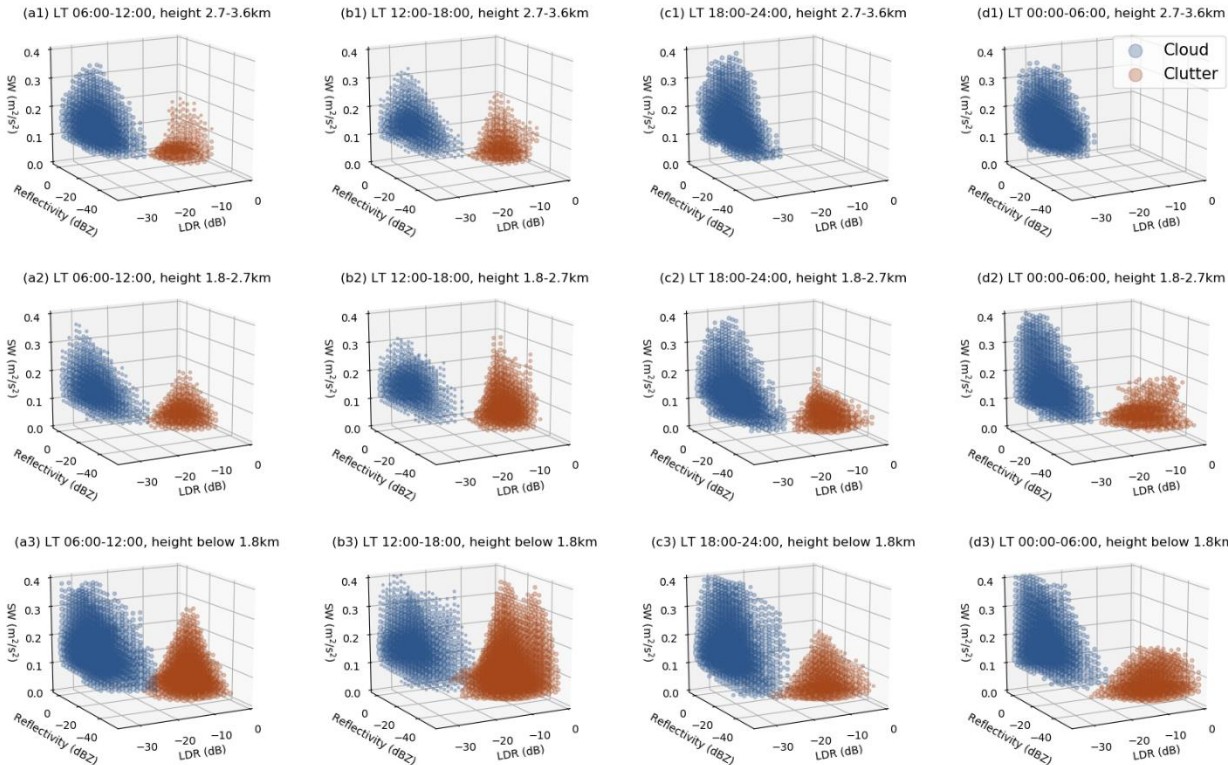

Figure 6. Same as Figure 5, except for cold season (October to March).



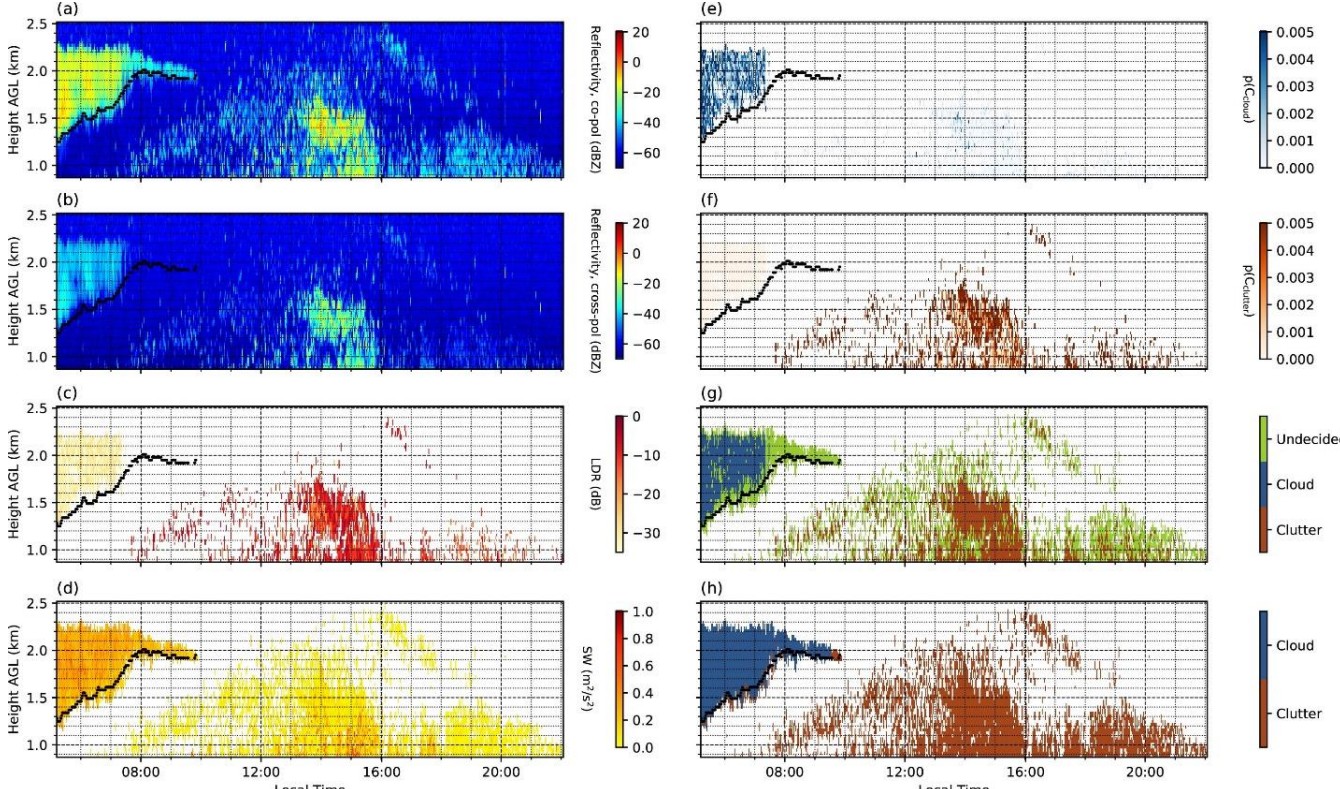


Figure 7. (a) Reflectivity of co-pol, (b) reflectivity of cross-pol, (c) LDR, (d) SW, (e) estimated probability of cloud, (f) estimated probability of clutter, (g) classification mask using Bayesian method and (h) classification mask after the spatial filter from local time 05:00 to 22:00 on September 24th, 2013. The black dots represent lidar detected cloud base height.





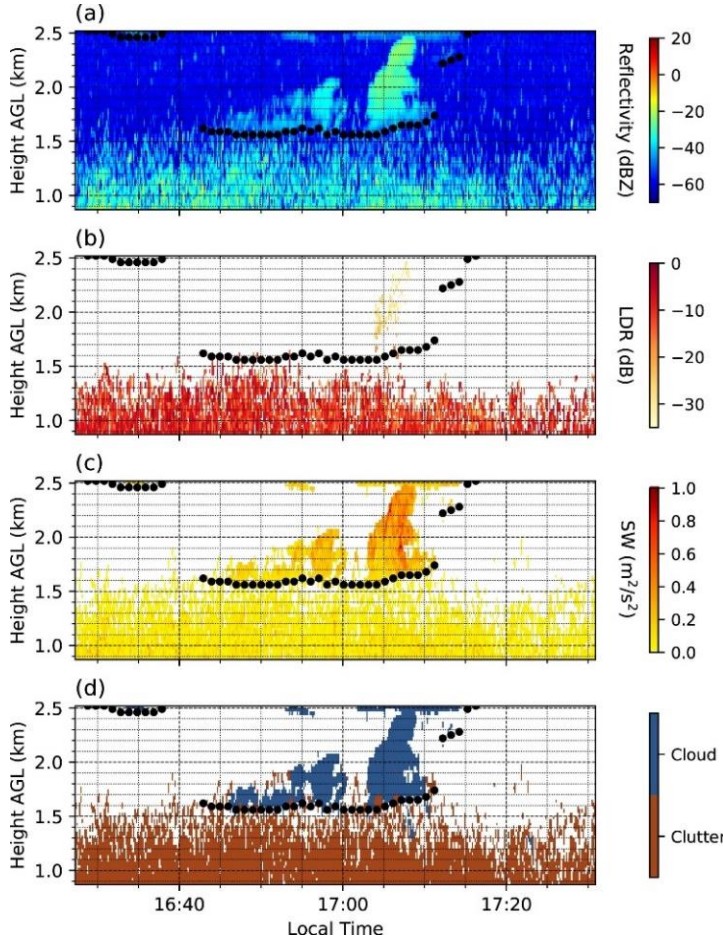

Figure 8. (a) Reflectivity, (b) LDR, (c) SW and (d) classification mask from local time 16:27 to 17:31 on April 15[th], 2014. The

black dots are lidar detected cloud base.





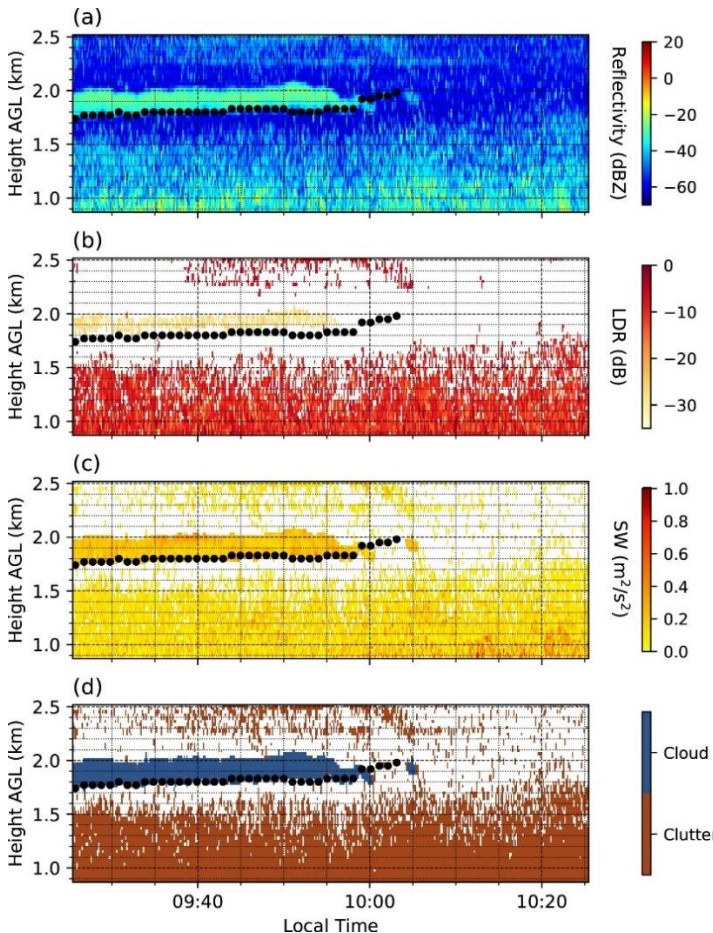

Figure 9. Same as Figure 8, except for local time 09:25 to 10:25 on October 12th, 2013.





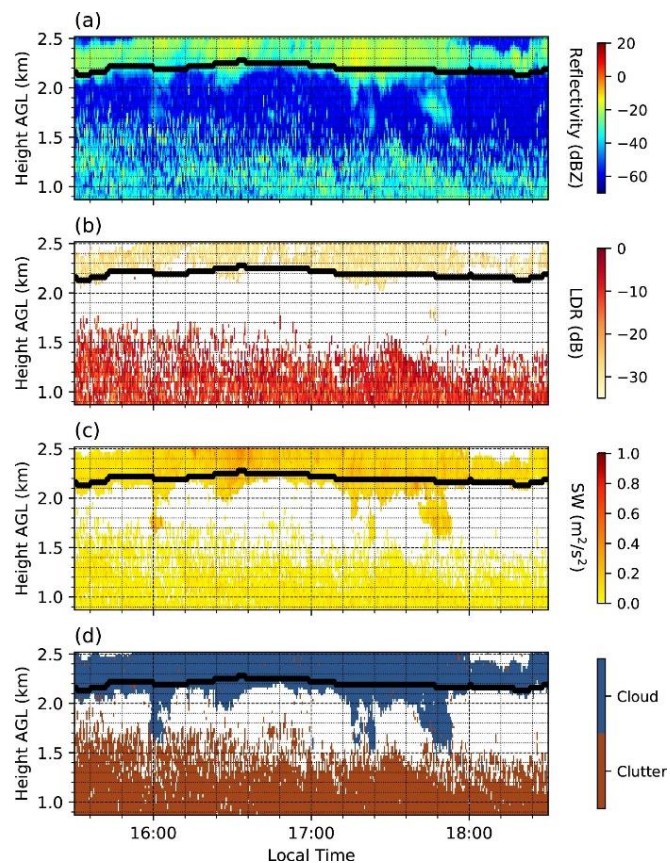


Figure 10. Same as Figure 8, except from local time15:30 to 18:30 on July 7th, 2014.





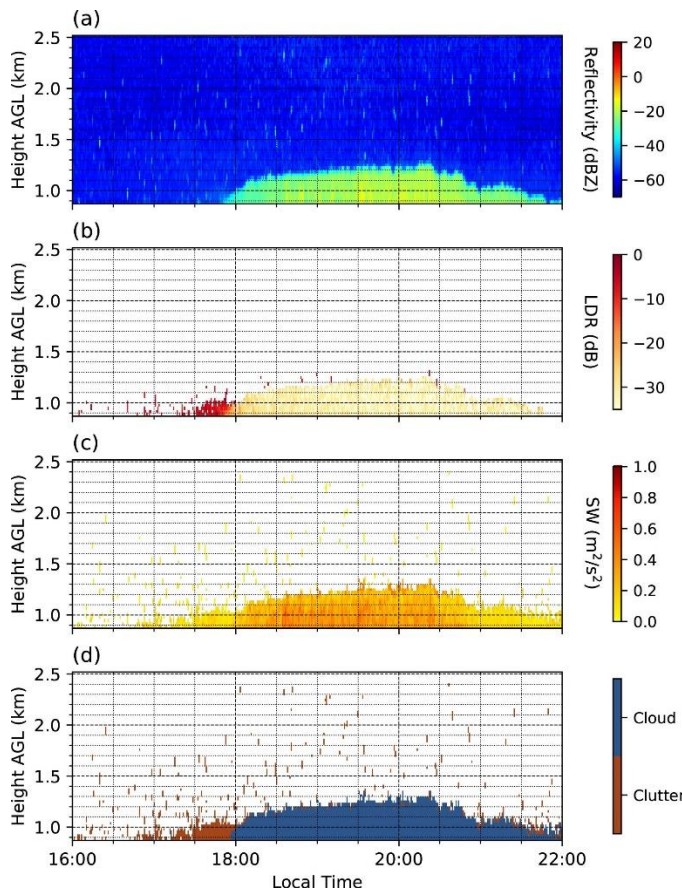

Figure 11. Same as Figure 8, except for local time 16:00 to 22:00 on February 4th, 2014.




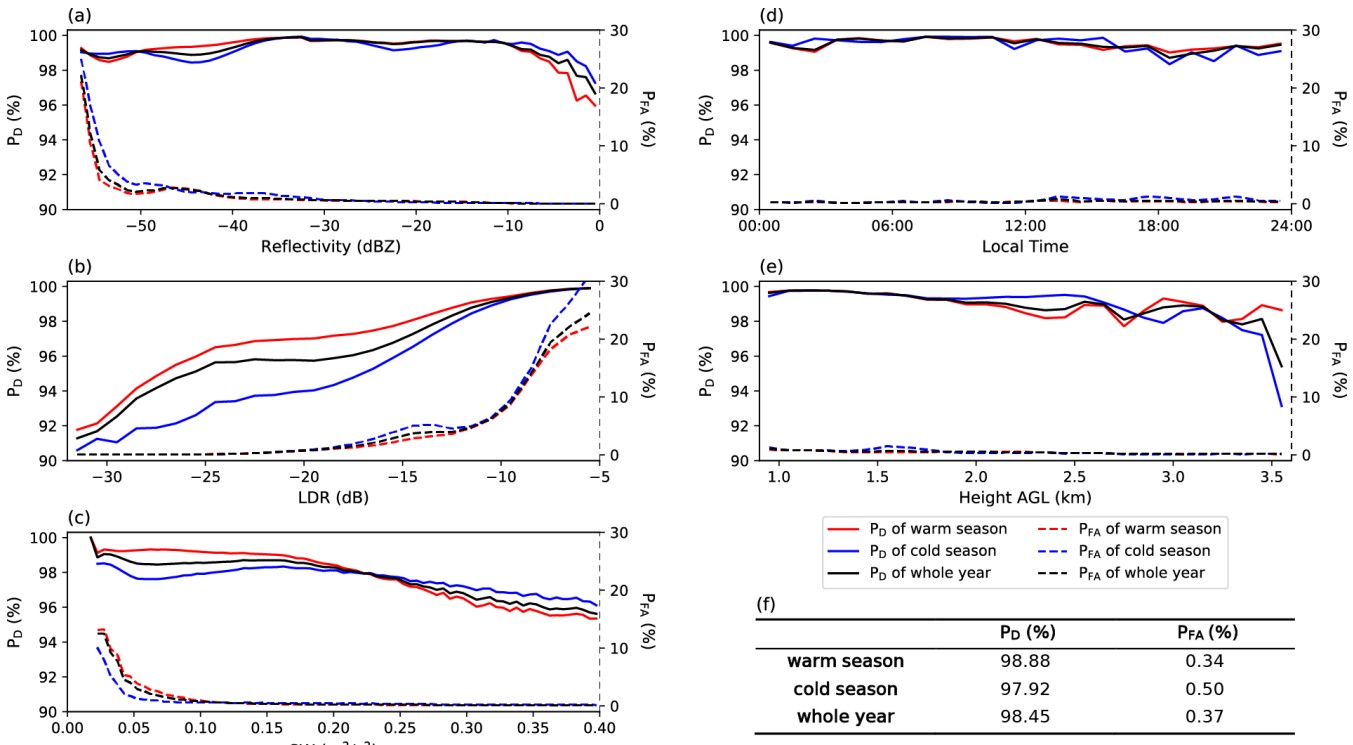

Figure 12. Probability of detection ($P_D$, solid line) and false alarm rate ($P_{FA}$, dashed line) as function of reflectivity (a), LDR

(b), SW (c), time (d) and height (e) for warm season (red line), cold season (blue line) and whole year (black line). The values

of $P_D$  and $P_{FA}$  for warm season, cold season and whole year are shown in (f).