# Peer review of "A robust low-level cloud and clutter discrimination method for ground-based millimeter-wavelength cloud radar"

_Atmospheric Measurement Techniques, 2020_

## Referee Comment (RC1) · Anonymous Referee #1 · 2 Nov 2020

Referee #1 Comments (RC): The article amt-2020-230 entitles 'A robust low-level cloud and clutter discrimination method for ground-based millimeter-wavelength cloud radar' by Xiaoyu Hu, Jinming Ge, et al., (2020).

General Comment: The research article mainly uses the measurements of ground-based 35-GHz cloud radar. The authors propose a clutter or biota discrimination method under the presence of low-level clouds. A Multi-dimensional PDF approach effectively has been utilized for Cloud and Clutter identification. Further, the obtained PDFs are used to train the AI-/ML-based Bayesian classifier for the classification mask generation. The scientific results and conclusions presented in a clear, concise, and

well-structured way with number and quality of figures/tables, appropriate use of the English language. However, one specific primary concern is as below:

Specific Major Concern:

Authors mainly showcase shallow layer clouds' persistently long presence whose flat horizontal cloud base was found using collocated ceilometers based cloud base observation (fig 8-10). Moreover, all the presented cloud and clutter discrimination cases have more apparent boundary discrimination between clutter and cloud, the meteorological target. Fig. 8 is a broken cumulus case, but the cloud base is elevated above 1.5 km, just above the clutter. Please include few shallow convective cloud cases surrounded by dense biota/insect clutter and cloud base having biota (near similar to authors fig 4, but here cloud is not weak because it possesses strong dBZ values above 10). The reason for asking it because It is interesting to see the performance of the proposed method under a dense clutter crowded the broken cumuli with duration may be less than a few minutes (e.g., see referred Luke et al., (2008) & Kalapureddy et al., AMT,(2018) within Fig 13-14 and A3).

For the better flow of the manuscript, immediately after Sec 3.3's Bayesian method, the method's functional performance can start directly with Sec. 3.4 by shifting the First Para in Pg.10 suitably after ending part of line number 217 (i.e.,'....near-surface in Figure 7c).'

All figures from 7-11 need to modify and extend the height (y-axis) up to 3.6 km as per multi-dimensional PDFs maximum height discussed with figures (5-6) and the probability of detecting height with figure 12e.

Page 12, Line 53 & Fig.10: How the precipitating drizzle droplets kept as clouds? Is it done manually or taken care of through the spatial filter? Because initially stated (at line no. 119), the non-cloud meteorological targets need to remove with the low-level atmosphere for the created PDFs' better accuracy to characterize cloud from clutters.

Minor technical corrections:

1. Pg2 Line no. 37: modify the sentence '…..of ground-based millimeter-wavelength cloud radars (MMCR) being………..'? This inclusion is the prerequisite for the Pg3 Line no. 48, an acronym MMCR? 2. Pg6 Ln114: '….the flat cloud boundary around 4.5 …' it is not flat but slant. 3. Pg7 Ln146 & (Fig 4e): '…..the height of maximum (Z' x V') up to 500 m above (below) the peak ….. ' In fact, it is five range gate spacing (of each 30 or 25 m) from the peak (see Fig 4e there are 11 dots between two red dots that is spread in 300 m height (3100-2800). Please check it? 4. Fig.4: (a) what is the reason not extending the identified black dots bottom and top portion of the melting layer before 20:24-20:27 LT (where high LDR at ~3 km with yellow backdrop seen)? 5. Fig.4 caption needs to recheck, especially correctness with 2nd line '…..Black dots and gray shading area in the left panels…..' unable to locate the gray shading in left panel except with (b) due to noise (may be removed by the NER with dBZ). 6. Pg11 Ln225: Please provide clarity on the used '…spatial filter with 'five range bins in vertical' do you mean with respect to 'height' and 'five range bins in the horizontal' do you mean 'concerning time'?? 7. Pg11 Ln239 end: delete 'is also'. 8. Pg 12 Ln250 & 267: it is confusing to read …lower SW values (below 0.4 m2 s-2) & higher SW values (around 0.3 m2 s-2). 9. Typo with Equation 4 where ',' read as FN'. 10. Pg 13 Ln283: modify sentence for completeness '……small portions of the data that overlaps.' 11. Pg 13 Ln289: I agree with the statement '….less fluctuating with time (Fig. 12d) and ' but not for height (Fig.12e) where PD shows significant changes above 2 km, especially in the cold season after 3.2 km altitude. 12. Fig. 11 caption should have mentioned the reason for missing the lidar cloud base.

Please also note the supplement to this comment:
https://amt.copernicus.org/preprints/amt-2020-230/amt-2020-230-RC1-supplement.pdf

---

## Referee Comment (RC2) · Anonymous Referee #3 · 15 Dec 2020

The authors developed a cloud and clutter discrimination algorithm for a ground-based millimeter-wave cloud radar system collocated to an MPL. The methodology to separate cloud from clutter is based on multivariate histograms that are used in a Bayes classification approach to provide categorical separation. Spectral width (SW), reflectivity, and linear depolarization ratio (LDR) are used to create joint histograms for cloud and insect clutter. The methodology is tested with a few case studies including shallow cumulus in the warm and cold seasons, uniform stratus embedded within insect layers, and precipitating stratocumulus. Comparisons are made to the MPL cloud base and show generally good agreement in the case studies. The approach is extended to one year of data and a probability of detection of 98% is obtained.

[Figure]

The methods, approach, and use of data all appear sound and the manuscript is organized well. The use of English could be improved in places. The novelty of the methods used in this manuscript should be more clearly called out when compared to previous works. These comments should be considered minor in scope, however.

Detailed comments:

Overall the manuscript could use a thorough edit for the use of English

One example is the use of 'clutters' rather than 'clutter'

In the Introduction, some clearer description of how this approach follows from, or is different from previous literature, should be added. It appears similar approaches exist in the literature but perhaps in pieces. For instance, insect detection with KAZRs may be better handled in spectra domain as by [1, 4], and LDR statistics with [2], and a similar but more comprehensive dual pol approach in [3] for scanning radars. Generally, LDR based estimates are widely used in the field as well.

[1] Luke, E. P., P. Kollias, K. L. Johnson, E. E. Clothiaux, A Technique for the Automatic Detection of Insect Clutter in Cloud Radar Returns. J. Atmos. Oceanic Technol. 25, 1498-1513, doi:10.1175/2007JTECHA953.1 (2008). (this is already cited) [2] Martner, B. E., and Moran, K. P. (2001), Using cloud radar polarization measurements to evaluate stratus cloud and insect echoes, J. Geophys. Res., 106( D5), 4891– 4897, doi:10.1029/2000JD900623. (not cited) [3] M. A. Rico-Ramirez and I. D. Cluckie, "Classification of Ground Clutter and Anomalous Propagation Using Dual-Polarization Weather Radar," in IEEE Transactions on Geoscience and Remote Sensing, vol. 46, no. 7, pp. 1892-1904, July 2008, doi: 10.1109/TGRS.2008.916979. (not cited) [4] Williams, C. R., Maahn, M., Hardin, J. C., & de Boer, G. (2018). Clutter mitigation, multiple peaks, and high-order spectral moments in 35 GHz vertically pointing radar velocity spectra. (not cited)

Lines 90-91 are repetitive

[Figure]

Lines 97-100, it appears the entire basis for the cloud and clutter histograms derives from the use of the MPL cloud base product. Are there other discriminants? How these histograms were obtained should be clearer. Furthermore, how do aerosols (e.g., dust) impact the histograms? Is there any dust in the case studies shown, and would the authors expect dust to hinder the discrimination of clouds and clutter in the algorithm itself?

Line 157, not sure if 'discrepant' is the right word

Lines 173-174, while the literature describes the number density and height of insects are temperature-dependent, do the species of insects themselves differ with season? Could a seasonal species dependence of insects have some bearing on the characteristics of the pdfs?

---

## Author Comment (AC2) · 22 Dec 2020

**Response to Anonymous Referee #3**

The authors developed a cloud and clutter discrimination algorithm for a ground-based millimeter-wave cloud radar system collocated to an MPL. The methodology to separate cloud from clutter is based on multivariate histograms that are used in a Bayes classification approach to provide categorical separation. Spectral width (SW), reflectivity, and linear depolarization ratio (LDR) are used to create joint histograms for cloud and insect clutter. The methodology is tested with a few case studies including shallow cumulus in the warm and cold seasons, uniform stratus embedded within insect layers, and precipitating stratocumulus. Comparisons are made to the MPL cloud base and show generally good agreement in the case studies. The approach is extended to one year of data and a probability of detection of 98% is obtained.

The methods, approach, and use of data all appear sound and the manuscript is organized well. The use of English could be improved in places. The novelty of the methods used in this manuscript should be more clearly called out when compared to previous works. These comments should be considered minor in scope, however.

**Response:** We thank the reviewer very much for his/her positive comments and suggestions on this manuscript. We have carefully read through the manuscript and corrected some grammar errors, including those pointed out by the reviewer. The novelty compared to previous works has also been described more clearly in the revised manuscript.

**Detailed comments:**

Overall the manuscript could use a thorough edit for the use of English. One example is the use of 'clutters' rather than 'clutter'

**Response:** We have carefully edited the use of English in the manuscript, including changing "clutter" into "clutters".

In the Introduction, some clearer description of how this approach follows from, or is different from previous literature, should be added. It appears similar approaches exist in the literature but perhaps in pieces. For instance, insect detection with KAZRs may be better handled in spectra domain as by [1, 4], and LDR statistics with [2], and a similar but more comprehensive dual pol approach in [3] for scanning radars. Generally, LDR based estimates are widely used in the field as well.

[1] Luke, E. P., P. Kollias, K. L. Johnson, E. E. Clothiaux, A Technique for the Automatic Detection of Insect Clutter in Cloud Radar Returns. J. Atmos. Oceanic Technol. 25, 1498-1513, doi:10.1175/2007JTECHA953.1 (2008). (this is already cited); [2] Martner, B. E., and Moran, K. P. (2001), Using cloud radar polarization measurements to evaluate stratus cloud and insect echoes, J. Geophys. Res., 106( D5), 4891–4897, doi:10.1029/2000JD900623. (not cited); [3] M. A. Rico-Ramirez and I. D. Cluckie, "Classification of Ground Clutter and Anomalous Propagation Using Dual-Polarization Weather Radar," in IEEE Transactions on Geoscience and Remote Sensing, vol. 46, no. 7, pp. 1892-1904, July 2008, doi: 10.1109/TGRS.2008.916979. (not cited); [4] Williams, C. R., Maahn, M., Hardin, J. C., & de Boer, G. (2018). Clutter mitigation, multiple peaks, and high-order spectral moments in 35 GHz vertically pointing radar velocity spectra. (not cited)

**Response:** We thank the reviewer for providing these relevant references, and we have cited these in the revised manuscript. We agree that the insect detection with KAZR is well handled in spectra domain as by Luke et al. (2008) and Williams et al. (2018). We think some other methods still have scientific significance, like the TEST algorithm proposed by Kalapureddy et al. (2018), which uses reflectivity measurements to characterize irregular echoes associated with clutter returns. Such methods do not require huge spectral data and the analysis processes are relatively simpler. The LDR statistic methods, such as proposed by Martner and Moran (2001) and Rico-Ramirez and Cluckie (2008), are for sure widely used in the field, but they can only be applied when both coand cross-polarized reflectivities are available, which may be not the case at low signal-to-noise ratio conditions as mentioned by Luke et al. (2008). That is why we use LDR statistics to create the PDFs and use a spatial filter to deal with these range gates when LDR measurements are unavailable. We have added these descriptions in the revised manuscript.

**Lines 90-91 are repetitive**

**Response:** We have deleted the second half of this sentence.

Lines 97-100, it appears the entire basis for the cloud and clutter histograms derives from the use of the MPL cloud base product. Are there other discriminants? How these histograms were obtained should be clearer. Furthermore, how do aerosols (e.g., dust) impact the histograms? Is there any dust in the case studies shown, and would the authors expect dust to hinder the discrimination of clouds and clutter in the algorithm itself?

**Response:** Yes, the histograms derived from the MPL cloud base product are the basis for the cloud and clutter separation. There is no other discriminant. The histograms are derived through the following steps: (1) we first collect all the reflectivity, LDR, SW data for cloud and clutter at different height and season based on MPL cloud base product; (2) we then divide all the samples into 12 panels according to their time and height ranges for warm and cold seasons separately (as shown in Fig. 5 and 6); (3) in each panel, the probability is calculated by  $p(\mathbf{X} = \mathbf{X}^0 | C_i) = n(\mathbf{X} = \mathbf{X}^0 | C_i) / \sum n(\mathbf{X} = \mathbf{X}^0 | C_i)$ , where  $p(\mathbf{X} = \mathbf{X}^0 | C_i)$  is the conditional probability of discriminants being  $\mathbf{X}^0$  for class  $C_i$ ,  $n(\mathbf{X} = \mathbf{X}^0 | C_i)$  is the number of samples of discriminants being  $\mathbf{X}^0$  for class  $C_i$ ,  $\sum n(\mathbf{X} = \mathbf{X}^0 | C_i)$  is the number of Fig. 5, as "The size of dots represents the value of probability density, which is calculated as the number of samples in each bin for each class (cloud or clutter) divided by the total number of samples in each bin for all classes."

For the impact of aerosols on the histogram, since MPL is not susceptible to the clutters, we use its cloud base product to separate cloud and clutter samples. All the non-cloud features identified from MPL, which are measured as significant echoes by KAZR, are considered as clutters, including insects, dust aerosols, pollen, or dry leaves. In other words, the clutter type is not the main concern of this study. Here, the MPL cloud base is derived from a feature detection using continuous wavelet transform analyses (Xie et al., 2017) that can well separate cloud and dust aerosols. Based on our current algorithm, we can not identify the clutter type (insect or dust), so we are not sure if there is any dust shown in the case studies. However, we do not expect that the dust would hinder the discrimination of clouds and clutters.

Line 157, not sure if 'discrepant' is the right word

**Response:** We have changed it to "distinct"

Lines 173-174, while the literature describes the number density and height of insects are temperature-dependent, do the species of insects themselves differ with season? Could a seasonal species dependence of insects have some bearing on the characteristics of the pdfs?

**Response:** Thanks for this interesting comment. Yes, the insect migration can cause seasonal variation of insect species. Note that cotton bollworm emerging in the far northeast of China would migrate into northern China in autumn, changing the local species of insect (Feng et al., 2007). The species of insects do affect the radar observation, due to their various shape, length, and wingbeat frequency, generating different morphology on spectra domain (Wang et al., 2017). However, such difference normally doesn't affect the reflectivity, LDR or SW (Wainwright et al., 2020), or the created PDF, consequently. Despite that, the difference of radar measurements between various insect species is smaller than that between insects and cloud droplets. Thus, we ignore the seasonal variation of insect species which may have a very small impact on the created PDFs.

**Reference:**

- Feng, H. Q., Wu, K. M., Ni, Y. X., Cheng, D. F., and Guo, Y. Y.: Return migration of Helicoverpa armigera (Lepidoptera: Noctuidae) during autumn in northern China, Bulletin of Entomological Research, 95, 361-370, 10.1079/ber2005367, 2007.
- Kalapureddy, M. C. R., Sukanya, P., Das, S. K., Deshpande, S. M., Pandithurai, G., Pazamany, A. L., Ambuj K, J., Chakravarty, K., Kalekar, P., Devisetty, H. K., and Annam, S.: A simple biota removal algorithm for 35 GHz cloud radar measurements, Atmospheric Measurement Techniques, 11, 1417-1436, 10.5194/amt-11-1417-2018, 2018.
- Wainwright, C. E., Reynolds, D. R., and Reynolds, A. M.: Linking Small-Scale Flight Manoeuvers and Density Profiles to the Vertical Movement of Insects in the Nocturnal Stable Boundary Layer, Scientific Reports, 10, 10.1038/s41598-020-57779-0, 2020.
- Wang, R., Hu, C., Fu, X., Long, T., and Zeng, T.: Micro-Doppler measurement of insect wing-beat frequencies with W-band coherent radar, Scientific Reports, 7, 10.1038/s41598-017-01616-4, 2017.
- Xie, H., Zhou, T., Fu, Q., Huang, J., Huang, Z., Bi, J., Shi, J., Zhang, B., and Ge, J.: Automated detection of cloud and aerosol features with SACOL micro-pulse lidar in northwest China, Optics Express, 25, 10.1364/oe.25.030732, 2017.

---

## Author Response (AR1)

**Response to Anonymous Referee #1**

*The article amt-2020-230 entitles 'A robust low-level cloud and clutter discrimination method for ground-based millimeter-wavelength cloud radar' by Xiaoyu Hu, Jinming Ge, et al., (2020).*
*General Comment:*
*The research article mainly uses the measurements of groundbased 35-GHz cloud radar. The authors propose a clutter or biota discrimination method under the presence of low-level clouds. A Multi-dimensional PDF approach effectively has been utilized for Cloud and Clutter identification. Further, the obtained PDFs are used to train the AI-/ML-based Bayesian classifier for the classification mask generation. The scientific results and conclusions presented in a clear, concise, and well-structured way with number and quality of figures/tables, appropriate use of the English language. However, one specific primary concern is as below:*

**Response:** We thank the reviewer very much for his/her positive comments and suggestions on this manuscript, which are very helpful for improving the quality of our paper. Our detailed responses to the comments are listed below.

*Specific Major Concern:*
*Authors mainly showcase shallow layer clouds' persistently long presence whose flat horizontal cloud base was found using collocated ceilometers based cloud base observation (fig 8-10). Moreover, all the presented cloud and clutter discrimination cases have more apparent boundary discrimination between clutter and cloud, the meteorological target. Fig. 8 is a broken cumulus case, but the cloud base is elevated above 1.5 km, just above the clutter. Please include few shallow convective cloud cases surrounded by dense biota/insect clutter and cloud base having biota (near similar to authors fig 4, but here cloud is not weak because it possesses strong dBZ values above 10). The reason for asking it because it is interesting to see the performance of the proposed method under a dense clutter crowded the broken cumuli with duration may be less than a few minutes (e.g., see referred Luke et al., (2008) & Kalapureddy et al., AMT,(2018) within Fig 13-14 and A3).*

**Response:** Thanks for this important comment. We have added another two cases in the

manuscript following the reviewer's suggestion. One is shallow convective cloud cases with clutter near cloud base in Fig. 1, another is a case of low-level clouds surrounded by dense clutters as shown in Fig. 2 below. These two cases show that the method can successfully discriminate most shallow convective cloud from dense clutter, although a few cloud droplets are misclassified as clutters. We have added the description for these two more cases and marked them as Fig. 11 and 12 in the revised manuscript.

[Figure]

Figure 1. (Figure 11 in the revised manuscript). Same as Figure 8 in the original manuscript, except from local time 04:18 to 05:45 on July 20th, 2014.

Figure 1 shows a case of broken cumulus and shallow convective clouds under stratus. One can see a few thin clouds (less than 300 m) below 1.5 km AGL during 04:30 to 05:10 and some broken cumulus from 04:30 to 04:50, which are like the case shown in Figure 8 but with lower cloud top and base heights ("more deeply buried" in the clutter layer). There may be many insets in the cloud, causing the large radar observed LDR, e.g., from 04:30 to 04:40 (greater than −15 dB, Fig. 1b), therefore, these range gates are classified as clutter by our algorithm (Fig. 1d). The clouds, where

are less effected by insects from 04:40 to 04:50 (lower LDR than −15 dB and higher SW than 0.4 m$^2$/s$^2$), are identified as cloud no doubt. Note that the occurrence of interlaced blocky appearance of classification masks around 04:40 (Fig. 1d). There are only little available LDR range gates there (Fig. 1b), meaning the classification masks are mostly contributed by the spatial filter (Sect. 3.4), which causes some misclassification (e.g., from 04:30 to 04:40) because the spatial correlation of clouds is reduced since they are largely contaminated by clutter. During 04:55 to 05:15, a few broken clouds higher away from the clutter layer are successfully identified by the algorithm, which is in accordance with the MPL lidar detections, indicating the spatial filter does work well when clouds are not adjacent to falsely identified masks. The shallow convective clouds after 05:15 are more turbulent (SW greater than 0.6 m$^2$/s$^2$, Fig. 1c) than these broken cumuli, thus are effectively identified as cloud even with dense clutter layer below. We believe the identified cloud mask below lidar cloud base from 05:15 to 05:30 are drizzle particles because of the virga reflectivity during that time (Fig. 1a).

[Figure]

Figure 2. (Figure 12 in the revised manuscript). Same as Figure 8 in the original manuscript, except from local time 14:15 to 17:00 on August 19th, 2013.

Figure 2 shows a case of low-level clouds completely surrounded by intense insets. This is the most difficult case to discriminate cloud and clutter, because cloud signals are heavily contaminated by clutters. Figure. 2d shows that the identified cloud masks correspond well with lidar cloud base during 14:15 to 16:00, due to lower LDR (less than −15 dB, Fig. 2b) and higher SW (greater than 0.4 $m^2/s^2$, Fig. 2c) of the cloud particles. However, the algorithm misses some clouds with low SW (around 0.2 $m^2/s^2$, Fig. 2c) during 16:00 to 16:40. Note that large amount of LDRs are unavailable for this cloud (Fig. 2b) and its structure is loose (Fig. 2a), especially around cloud edges where clutter signals are even stronger than cloud. In this circumstance, the algorithm can only identify part of the cloud.

*For the better flow of the manuscript, immediately after Sec 3.3's Bayesian method, the method's functional performance can start directly with Sec. 3.4 by shifting the First Para in Pg.10 suitably after ending part of line number 217 (i.e.,'....near-surface in Figure 7c).'*

**Response:** We have modified this paragraph in Sect. 3.4.

*All figures from 7-11 need to modify and extend the height (y-axis) up to 3.6 km as per multi-dimensional PDFs maximum height discussed with figures (5-6) and the probability of detecting height with figure 12e.*

**Response:** We modified the height scale in Figures 7-11 up to 3.6 km as shown below (Figs. 3-7) and can see large blank areas above 3 km, which might reduce the readability for these figures. Thus, we believe the original figures are much clear for the details. In addition, the probabilities of detection ($P_D$) and false alarm rate ($P_{FA}$) as a function of height are shown in Fig. 12e. That gives info for the reader to see the algorithm's performance with height. So, we kept the original figures unchanged in the revised version.

[Figure]

Figure 3. Original figure 7 in the manuscript (top panel within solid line box), and modified figure with extending y-axis (bottom panel within dashed line box)

[Figure]

Figure 4. Original figure 8 in the manuscript (left panel), and modified figure with extending y-axis (right panel)

[Figure]

Figure 5. Original figure 9 in the manuscript (left panel), and modified figure with extending y-axis (right panel)

[Figure]

Figure 6. Original figure 10 in the manuscript (left panel), and modified figure with extending y-axis (right panel)

[Figure]

Figure 7. Original figure 11 in the manuscript (left panel), and modified figure with extending y-axis (right panel)

*Page 12, Line 53 & Fig.10: How the precipitating drizzle droplets kept as clouds? Is it done manually or taken care of through the spatial filter? Because initially stated (at line no. 119), the non-cloud meteorological targets need to remove with the low-level atmosphere for the created*

*PDFs' better accuracy to characterize cloud from clutters.*

**Response:** The precipitating drizzle droplets are kept as clouds by the method. Yes, we stated that the non-cloud meteorological targets, including precipitation, melting layer and drizzle, are removed before creating PDF from the training data. However, note that both precipitation and melting layer are merely identified by radar, while drizzle is distinguished from the MPL derived cloud base. That means we can identify precipitation and melting layer before the process of separating cloud from clutter by using the test data, but cannot classify drizzle only using the KAZR observation. Thus, the drizzle is kept as cloud by the method. We think it is acceptable because clutter can be removed on which we mainly focused, and preserving drizzle as cloud rather than clutter, like Luke et al. (2008) explained.

*Minor technical corrections:*
*1. Pg2 Line no. 37: modify the sentence '.....of ground-based millimeter-wavelength cloud radars (MMCR) being...........'? This inclusion is the prerequisite for the Pg3 Line no. 48, an acronym MMCR?*

**Response:** This has been corrected.

*2. Pg6 Ln114: '....the flat cloud boundary around 4.5 ...' it is not flat but slant.*

**Response:** We have changed "flat" to "slanted".

*3. Pg7 Ln146 & (Fig 4e): '.....the height of maximum (Z' x V') up to 500 m above (below) the peak ..... ' In fact, it is five range gate spacing (of each 30 or 25 m) from the peak (see Fig 4e there are 11 dots between two red dots that is spread in 300 m height (3100-2800). Please check it?*

**Response:** You are right about the depth of the melting layer in the Fig. 4e (11 dots and 300 m). Here, by "500 m", we don't mean the real depth, but the "maximum" half depth of melting layer (top to middle, or middle to bottom), that is a threshold of melting layer in the identification algorithm, which was modified from Khanal et al. (2019). The main steps are:

a. Compute the reflectivity', velocity', |reflectivity'|×|velocity'|, and (|reflectivity'|×|velocity'|)". (The sign of absolute value was missed in the original manuscript and has been added now.)

b. Find the altitude with maximum |reflectivity'|×|velocity'|, which is the middle of melting layer.

c. Search for top of melting layer as maximum (|reflectivity'|×|velocity'|)" up to 500 m above middle of melting layer.

d. Search for bottom of melting layer as maximum (|reflectivity'|×|velocity'|)" up to 500 m below middle of melting layer.

We have changed some statements in the manuscript. It should be less confusing now.

*4. Fig.4: (a) what is the reason not extending the identified black dots bottom and top portion of the melting layer before 20:24-20:27 LT (where high LDR at ~3 km with yellow backdrop seen)?*

**Response:** It is true that there is still undetected melting layer around 20:24-20:27. This is because the melting layer identification algorithm only works for precipitation profile. No precipitation is detected during that time, so does the melting layer. We tried to develop an independent melting layer identification method from precipitation, but that caused some false detections. Since the undetected melting layer is quiet few by our manual scanning, we think it is acceptable to get the PDF.

*5. Fig.4 caption needs to recheck, especially correctness with 2nd line '.....Black dots and gray shading area in the left panels…..' unable to locate the gray shading in left panel except with (b) due to noise (may be removed by the NER with dBZ).*

**Response:** The precipitation is the shaded area after 20:27 and below 2.8 km, which may be not clear in the original manuscript. We modified the figure as shown below.

[Figure]

Figure 8 Modified Fig 4 in the original manuscript

*6. Pg11 Ln225: Please provide clarity on the used '...spatial filter with 'five range bins in vertical' do you mean with respect to 'height' and 'five range bins in the horizontal' do you mean 'concerning time'??*

**Response:** Yes. "Vertical" refers the height and "horizontal" is time. We have change it as "spatial filter with five range bins respecting to heigh (150 m) and five range bins concerning time (21.4 s)".

*7. Pg11 Ln239 end: delete 'is also'.*

**Response:** We think keeping the "is also" is more easily understandable. So, we kept the original sentence unchanged in the revised version.

*8. Pg 12 Ln250 & 267: it is confusing to read ...lower SW values (below 0.4 $m^2$ $s^{-2}$) & higher SW values (around 0.3 $m^2$ $s^{-2}$).*

**Response:** We have changed the "around 0.3 $m^2/s^2$" to "around 0.6 $m^2/s^2$ in the cumulus from 18:00 to 20:30" in Line 267. The original "0.3 $m^2/s^2$" is calculated from the whole identified cloud. However, the SW of cumulus during 18:00-20:30 is obviously higher than that during 20:30-22:00. We recalculate the SW during that time, it should be "around 0.6 $m^2/s^2$".

*9. Typo with Equation 4 where ',' read as FN'.*

**Response:** We have deleted the ",".

*10. Pg 13 Ln283: modify sentence for completeness '……small portions of the data that overlaps.'*

**Response:** We have rewritten this sentence as "……, however, which are only small portions of the whole data as shown in Figures 5 and 6".

*11. Pg 13 Ln289: I agree with the statement '….less fluctuating with time (Fig. 12d) and ' but not for height (Fig.12e) where $P_D$ shows significant changes above 2 km, especially in the cold season after 3.2 km altitude.*

**Response:** Yes, we have explained this in the end of the sentence (which is in the next page that may be missed). It was "except for $P_D$ above 3.2 km, where the clutter is extremely rare (fewer samples)". We changed the original number of "3.5 km" to "3.2 km".

*12. Fig. 11 caption should have mentioned reason for missing lidar cloud base.*

**Response:** The lidar observation was not available that day. However, the different LDR values before and after 18:00 help to distinguish cloud from clutter reliably. We have added the expiation in the caption.


**Response to Anonymous Referee #3**

*The authors developed a cloud and clutter discrimination algorithm for a ground-based millimeter-wave cloud radar system collocated to an MPL. The methodology to separate cloud from clutter is based on multivariate histograms that are used in a Bayes classification approach to provide categorical separation. Spectral width (SW), reflectivity, and linear depolarization ratio (LDR) are used to create joint histograms for cloud and insect clutter. The methodology is tested with a few case studies including shallow cumulus in the warm and cold seasons, uniform stratus embedded within insect layers, and precipitating stratocumulus. Comparisons are made to the MPL cloud base and show generally good agreement in the case studies. The approach is extended to one year of data and a probability of detection of 98% is obtained.*

*The methods, approach, and use of data all appear sound and the manuscript is organized well. The use of English could be improved in places. The novelty of the methods used in this manuscript should be more clearly called out when compared to previous works. These comments should be considered minor in scope, however.*

**Response:** We thank the reviewer very much for his/her positive comments and suggestions on this manuscript. We have carefully read through the manuscript and corrected some grammar errors, including those pointed out by the reviewer. The novelty compared to previous works has also been described more clearly in the revised manuscript.

*Detailed comments:*

*Overall the manuscript could use a thorough edit for the use of English. One example is the use of 'clutters' rather than 'clutter'*

**Response:** We have carefully edited the use of English in the manuscript, including changing "clutter" into "clutters".

*In the Introduction, some clearer description of how this approach follows from, or is different from previous literature, should be added. It appears similar approaches exist in the literature but perhaps in pieces. For instance, insect detection with KAZRs may be better handled in spectra*

*domain as by [1, 4], and LDR statistics with [2], and a similar but more comprehensive dual pol approach in [3] for scanning radars. Generally, LDR based estimates are widely used in the field as well.*

*[1] Luke, E. P., P. Kollias, K. L. Johnson, E. E. Clothiaux, A Technique for the Automatic Detection of Insect Clutter in Cloud Radar Returns. J. Atmos. Oceanic Technol. 25, 1498-1513, doi:10.1175/2007JTECHA953.1 (2008). (this is already cited); [2] Martner, B. E., and Moran, K. P. (2001), Using cloud radar polarization measurements to evaluate stratus cloud and insect echoes, J. Geophys. Res., 106( D5), 4891–4897, doi:10.1029/2000JD900623. (not cited); [3] M. A. Rico-Ramirez and I. D. Cluckie, "Classification of Ground Clutter and Anomalous Propagation Using Dual-Polarization Weather Radar," in IEEE Transactions on Geoscience and Remote Sensing, vol. 46, no. 7, pp. 1892-1904, July 2008, doi: 10.1109/TGRS.2008.916979. (not cited); [4] Williams, C. R., Maahn, M., Hardin, J. C., & de Boer, G. (2018). Clutter mitigation, multiple peaks, and high-order spectral moments in 35 GHz vertically pointing radar velocity spectra. (not cited)*

**Response:** We thank the reviewer for providing these relevant references, and we have cited these in the revised manuscript. We agree that the insect detection with KAZR is well handled in spectra domain as by Luke et al. (2008) and Williams et al. (2018). We think some other methods still have scientific significance, like the TEST algorithm proposed by Kalapureddy et al. (2018), which uses reflectivity measurements to characterize irregular echoes associated with clutter returns. Such methods do not require huge spectral data and the analysis processes are relatively simpler. The LDR statistic methods, such as proposed by Martner and Moran (2001) and Rico-Ramirez and Cluckie (2008), are for sure widely used in the field, but they can only be applied when both co- and cross-polarized reflectivities are available, which may be not the case at low signal-to-noise ratio conditions as mentioned by Luke et al. (2008). That is why we use LDR statistics to create the PDFs and use a spatial filter to deal with these range gates when LDR measurements are unavailable. We have added these descriptions in the revised manuscript.

*Lines 90-91 are repetitive*

**Response:** We have deleted the second half of this sentence.

*Lines 97-100, it appears the entire basis for the cloud and clutter histograms derives from the use of the MPL cloud base product. Are there other discriminants? How these histograms were obtained should be clearer. Furthermore, how do aerosols (e.g., dust) impact the histograms? Is there any dust in the case studies shown, and would the authors expect dust to hinder the discrimination of clouds and clutter in the algorithm itself?*

**Response:** Yes, the histograms derived from the MPL cloud base product are the basis for the cloud and clutter separation. There is no other discriminant. The histograms are derived through the following steps: (1) we first collect all the reflectivity, LDR, SW data for cloud and clutter at different height and season based on MPL cloud base product; (2) we then divide all the samples into 12 panels according to their time and height ranges for warm and cold seasons separately (as shown in Fig. 5 and 6); (3) in each panel, the probability is calculated by $p(X = X^O|C_i) = n(X = X^O|C_i)/\sum n(X = X^O|C_i)$ , where $p(X = X^O|C_i)$ is the conditional probability of discriminants being $X^O$ for class $C_i$, $n(X = X^O|C_i)$ is the number of samples of discriminants being $X^O$ for class $C_i$, $\sum n(X = X^O|C_i)$ is the number of discriminant samples being $X^O$ for all classes. We have added the details of calculation in Sect. 3.2 (Line 175) in the revised manuscript, as "……, which is calculated as the number of samples in each discriminant range for each class (clouds or clutters), divided by the total number of samples in each discriminant range for all classes."

For the impact of aerosols on the histogram, since MPL is not susceptible to the clutters, we use its cloud base product to separate cloud and clutter samples. All the non-cloud features identified from MPL, which are measured as significant echoes by KAZR, are considered as clutters, including insects, dust aerosols, pollen, or dry leaves. In other words, the clutter type is not the main concern of this study. Here, the MPL cloud base is derived from a feature detection using continuous wavelet transform analyses (Xie et al., 2017) that can well separate cloud and dust aerosols. Based on our current algorithm, we can not identify the clutter type (insect or dust), so we are not sure if there is any dust shown in the case studies. However, we do not expect that the dust would hinder the discrimination of clouds and clutters.

*Line 157, not sure if 'discrepant' is the right word*

**Response:** We have changed it to "distinct"

*Lines 173-174, while the literature describes the number density and height of insects are temperature-dependent, do the species of insects themselves differ with season? Could a seasonal species dependence of insects have some bearing on the characteristics of the pdfs?*

**Response:** Thanks for this interesting comment. Yes, the insect migration can cause seasonal variation of insect species. Note that cotton bollworm emerging in the far northeast of China would migrate into northern China in autumn, changing the local species of insect (Feng et al., 2007). The species of insects do affect the radar observation, due to their various shape, length, and wing-beat frequency, generating different morphology on spectra domain (Wang et al., 2017). However, such difference normally doesn't affect the reflectivity, LDR or SW (Wainwright et al., 2020), or the created PDF, consequently. Despite that, the difference of radar measurements between various insect species is smaller than that between insects and cloud droplets. Thus, we ignore the seasonal variation of insect species which may have a very small impact on the created PDFs.

[revised manuscript text omitted]